# FreqMAE: Frequency-Aware Masked Autoencoder for Multi-Modal IoT Sensing Applications

## ABSTRACT

This paper introduces FreqMAE, a novel self-supervised learning framework that synergizes masked autoencoding (MAE) with physics-informed signal insights to capture feature patterns from multi-modal IoT sensing signals. By enhancing the representation of sensor data semantics in a latent feature space, FreqMAE diminishes the dependence on data labeling and boosts the accuracy of downstream AI tasks. Unlike paradigms relying on data augmentations, such as contrastive learning, FreqMAE's automated approach avoids handcrafted label-invariant transformations. Adapting MAE for IoT sensing signals, we present three contributions from frequency domain insights: First, a Temporal-Shifting Transformer (TS-T) encoder that enables temporal interactions while distinguishing different frequency regions; Second, a factorized multi-modal fusion mechanism that leverages cross-modal correlations while allowing for modality-private features; Third, a hierarchically weighted loss function that prioritizes the reconstruction of important frequency components and high Signal-to-Noise Ratio (SNR) samples. Comprehensive evaluations on two sensing applications validate FreqMAE's proficiency in reducing labeling needs and enhancing resilience against domain shifts.

## 1 INTRODUCTION

The paper advances the state of the art in self-supervised learning from time-series sensor data. Self-supervised learning aims to transform *unlabeled* input data into a latent space that captures data semantics, simplifying extensive downstream tasks. Two popular ways to do so are *contrastive learning* and *masked autoencoders*. Contrastive learning utilizes data augmentations, such as image rotations, that maintain content semantics. By comparing pairs of these semantically similar inputs against random pairs, neural networks are trained to cluster similar items in latent space. On the other hand, MAEs, which don't require designing semantics-preserving augmentations, conceal parts of the input and train the network to reconstruct these sections accurately. The insight behind MAEs is that accurate reconstruction of masked sections indicates the network's ability to discern higher-level semantics. For example, if the network can deduce an object's traits from partial data, it can likely reconstruct obscured sections of that object. With a latent space that effectively represents high-level object attributes, training subsequent inference tasks is more efficient. Hence, label-free MAEs optimize the training process for downstream AI tasks, achieving better accuracy even with limited data samples [33].

Although MAEs excelled in vision and natural language domains [20, 37, 64], their performance on time-series sensing signals has been inferior to contrastive frameworks [62]. We find that appropriately integrating insights from a conventional signal processing perspective can effectively simplify the optimization space and boost the performance of MAEs. Therefore, we introduce FreqMAE, a specialized MAE for multi-modal IoT sensing. It integrates three

distinct frequency-aware insights applicable across sensing tasks, which set FreqMAE apart from standard MAEs, tailoring it for time-frequency analysis.

First, we design a frequency-aware Transformer variant tailored for sensor spectrogram encoding. While Transformers [57] excel in handling complex data distributions due to their adaptive neural attention, using Vision Transformer (ViT) encoders directly on sensor spectrograms by treating them as images, has proven less effective [23]. This is because ViT encoders utilize global attention across all input areas, only suitable for visual data where semantics remain consistent irrespective of object position or transformation. Yet, for spectrogram data, translation and scaling of frequencies can significantly change the semantics of sensor measurements. Moreover, spectrogram amplitudes and fundamental frequencies exhibit gradual temporal shifts due to the non-stationary nature of physical elements [41]. Addressing these nuances, we present a *Temporal-Shifting Transformer (TS-T)* that separately handles frequency and time domains, aligning with time-series signal characteristics. In the frequency domain, we integrate a *local* attention mechanism that clusters and partitions the short-time Fourier windows of the spectrogram into localized windows. Conversely, we compute the attention with frequencies and their shifted harmonic components in the temporal domain. This temporal shift operation *preserves the spectral structure while representing shifting frequency behavior.*

Second, we introduce a factorized data fusion mechanism that emphasizes both cross-modal correlations and modality-private features. The insight here is that synchronized modalities not only share information from the same physical stimuli but also offer unique perspectives that complement each other through collaboration [69]. To extract comprehensive information, we apply single masking to the input of each modality. This results in two distinct feature spaces post-encoding: *(i) a private* and *(ii) a shared space.* The private space *captures distinct modality-specific patterns*, emphasizing self-reconstruction. The shared space, on the other hand, *captures cross-modal information*, where one modality's input is reconstructed using shared embeddings from other modalities. To achieve this, we utilize two specialized lightweight decoders, *ensuring no extra overhead during fine-tuning or inference.*

Third, we propose a hierarchically weighted loss function emphasizing important frequency regions and high Signal-to-Noise Ratio (SNR) samples. To illustrate the benefits of weighting, we consider IoT applications, where crucial information is predominantly found in the low-frequency components, whereas high-frequency sections are mostly noise [33]. Consequently, emphasizing the accurate reconstruction of these low-frequency parts during training bolsters the quality of representation learning. Moreover, high SNR measurements, with substantial energy content, provide accurate insights, enhancing representation learning's efficacy. For instance, in vehicle classification via audio and seismic sensors, measurements captured when vehicles are nearby are especially informative [63].

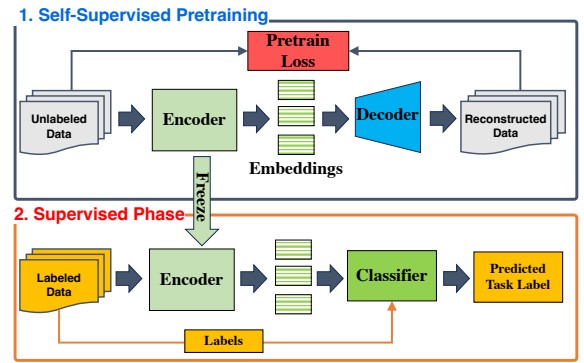

Figure 1: Masked Autoencoder (MAE) Workflow

This work is motivated by the rise in embedded device computational power, coupled with deep neural networks' (DNN) robust modeling, which propelled the Artificial Intelligence of Things (AIoT) domain, fostering advancements in activity detection, vehicle tracking, and smart healthcare [3, 22, 45, 50, 65]. Most existing work [12, 16, 27, 36, 66] relies heavily on *supervised* learning that requires substantial amounts of labeled data for training purposes. This reliance poses practical challenges, as time-series labels can only be collected in predefined controlled environments. Unlike the common practice of mass labeling image and text data through post-hoc crowdsourcing, *understanding sensing signals and obtaining their labels is not straightforward* [49]. Moreover, *DNN models trained on data from limited environments often exhibit sensitivity to unforeseen changes in the actual deployment setting* [60].

By utilizing *self-supervised learning*, we train the encoder *without the need for labeled data*. Subsequently, we perform supervised fine-tuning using a limited number of data labels to train the downstream inference task. This approach is highly label-efficient and yields pretrained data encoders with enhanced robustness against environmental variations. Unlike contrastive learning frameworks [7, 13] which heavily rely on human intuition to create label-invariant transformations, FreqMAE only employs simple random masking as the preprocessing step. It also integrates physical signal knowledge that is applicable across various sensing applications as improvements, resulting in *higher automaticity and extensibility*.

We extensively evaluate FreqMAE using four datasets, demonstrating its superior performance over existing approaches in various sensing applications and downstream tasks. The results highlight the exceptional potential of the self-supervised FreqMAE framework as a step towards building foundation models specially tailored for sensing streams and time series data. Beyond the dataset evaluations, we use a real-world case study to demonstrate the robustness of FreqMAE. One standout feature is its exceptional performance in the face of environmental variations. FreqMAE shows unparalleled capability in managing dynamic, real-life scenarios, affirming its utility for representing information from dynamic sensing streams.

The rest of this paper is organized as follows. Section 2 presents background knowledge used in this paper. Section 3 introduces FreqMAE design details. Section 4 provides the experiment details and results. Section 5 reviews the related work, and Section 6 discusses the limitations and concludes this paper.

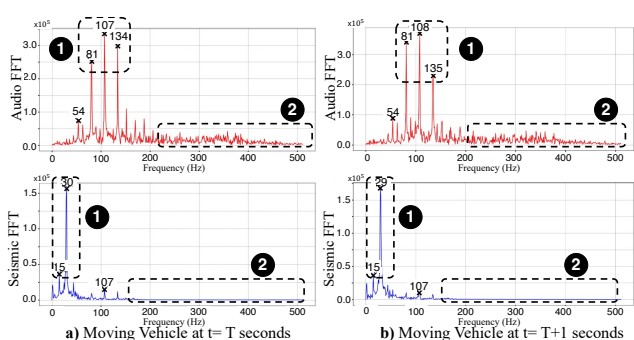

**a) Moving Vehicle at t= T seconds**   **b) Moving Vehicle at t= T+1 seconds**

Figure 2: Audio FFT signatures for a moving vehicle. ❶ The presence of characteristic peaks in localized regions needs local harmonic associations and shift-sensitive representations. ❷ Higher frequency regions mostly contain noise.

## 2 PRELIMINARIES

This section outlines the foundational concepts of self-supervised learning and the inspirations behind FreqMAE's design.

### 2.1 Masked Autoencoders

Compared to the prevalent contrastive learning paradigm for IoT data [11, 53, 59], which heavily relies on domain knowledge for designing label-invariant transformations (*i.e.*, augmentations), we introduce a fully automated self-supervised approach based on MAEs [20]. This approach, *free from augmentations*, applies broadly to many sensing contexts and *drastically reduces labeled data dependence*. Figure 1 illustrates the MAE structure, featuring an encoder, a decoder, and a downstream classifier, with a two-phase training: self-supervised pretraining and supervised fine-tuning.

The aim of pretraining is to leverage extensive unlabeled data for extracting versatile representations applicable to various downstream tasks. Specifically, we employ random masking on segments of the unlabeled spectrograms. The encoder then processes the masked data, creating a low-dimensional data embedding. The decoder's role is to reconstruct the masked regions using these encoded embeddings. The training aims to minimize the discrepancy between the decoded results and the original data within masked areas. To encourage the model to capture overarching semantics over low-level interpolations, we apply masking at the granularity of frequency patches with a high masking ratio.

In the fine-tuning stage, we discard the decoder and directly connect the encoder to a lightweight classifier (*i.e.*, one fully connected layer). During this phase, the pretrained encoder parameters remain fixed, and the linear classifier is updated using the representations generated by FreqMAE, which are based on limited labels specific to the downstream task. This approach offers two advantages: *(i) the need for fewer labels for convergence* [28] and *(ii) faster training*.

### 2.2 Characteristics of IoT Sensing Data

IoT sensing data exhibit unique characteristics that set them apart from other contexts. Following common practices [31, 66], we use spectrogram data after a short-time Fourier transform (STFT) on the raw input, as the modality input. We carefully examine the fundamental properties of such spectrograms to guide the design of FreqMAE. Figure 2 presents two sensor (audio and seismic) readings

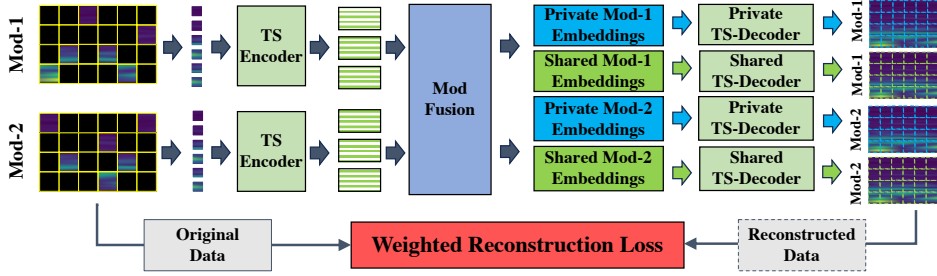

**Figure 3: FreqMAE design components with self-supervised pretraining workflow.**

from two consecutive time windows for a moving vehicle, collected as it passes by the sensors. Several observations are highlighted.

*2.2.1* **No Scale and Shift Invariance.** While vanilla MAE employs global attention due to visual objects' invariance to translation or scaling, this assumption doesn't hold for IoT data. Here, the positioning and scaling of frequency content significantly influence semantics. Thus, global self-attention might be less effective when time-frequency information is predominantly local. For instance, only linking harmonic patches vertically through frequency (see ❶ in Figure 2) may be suboptimal due to recurring harmonics while associating the shifted harmonics horizontally through time can yield more comprehensive insights into non-stationary patterns.

*2.2.2* **Multi-Modal Fusion.** IoT data stems from various sensors, such as accelerometers, gyroscopes, and magnetometers, each providing a distinct perspective into the observed event. By fusing information from multiple sensors, a richer understanding and increased system efficacy can be achieved [6]. Therefore, aligning with the emerging trend on multi-modal fusion [5, 30, 35, 47, 58, 66], an effective SSL framework should support data fusion across diverse modalities and feature generalization across various sensors.

*2.2.3* **Differentiated Frequency and Sample Importance.** Regarding the reconstruction objective in MAE, we observed that differentiated importance should be imposed locally among different frequency bands and globally among different samples. First, in physical sensing tasks, it is well-known that valuable information tends to be found in the low-frequency sections of the spectrogram [33]. Conversely, the very high-frequency sections often consist mostly of noise (*e.g.,* ❷ in Figure 2). Second, due to external factors and the nature of physical sensing data, some samples are more important than others regarding the detection of the observed phenomenon. For instance, samples with higher SNR provide more useful information than lower SNR samples that include noise.

## 3 FRAMEWORK

In this section, we introduce FreqMAE and its three novel components (motivated by the aforementioned characteristics).

### 3.1 Overview

The aim of FreqMAE is to generate representational embeddings for unlabeled time series sensing data from multiple collaborating sensory modalities. Assuming we have a collection of $P$ modalities $\mathcal{M} = \{M_1, M_2, \ldots, M_P\}$ and a large set of $N$ unlabeled training samples $\mathcal{X} = \{\mathbf{x}_1, \mathbf{x}_2, \ldots, \mathbf{x}_N\}$ from all modalities, where each

sample is a fixed-length signal window. Sample $\mathbf{x}_{ij}$ represents the input from sensory modality $M_j$ within sample $\mathbf{x}_i$. Then, the objective of FreqMAE can be formulated as: $\mathbf{h}_{ij} = E_j(\mathbf{x}_{ij})$, where $\mathcal{E} = \{E_1, E_2, \ldots, E_P\}$ are FreqMAE encoders for each modality and $\mathbf{h}_{ij}$ is the embedding vector sample representations of $\mathbf{x}_{ij}$. The original modal input forms a multivariate time series, which we transform via Short-Time Fourier Transform (STFT) for time-frequency representation (*i.e.,* spectrogram).

Figure 3 illustrates the FreqMAE pretraining process. We start by dividing time-frequency sample spectrograms into non-overlapping regular grid patches. These patches are then flattened and embedded through a linear projection. In line with previous work [36], we found no discernible advantage in incorporating positional embeddings (Analysis at Appendix D.5).

We then randomly mask out a large portion of spectrogram patches which is the key ingredient for efficient self-supervised pretraining [20]. In this process, the masking resembles a Bernoulli process, where each patch has a probability $p$ of being masked (also called the *masking ratio*). Since spectrograms provide a two-dimensional representation of time-frequency components, we explored both unstructured and structured masking strategies. Our investigation revealed that unstructured random masking delivers the best pretraining performance (analysis at Appendix D.2). Similar to images [20], a high masking rate, ranging from 70% to 80%, is most conducive to representation learning.

FreqMAE utilizes Temporal-Shifting (TS) Transformer encoders for each modality, a transformer design incorporating localized attention with a spectrogram-compatible shifting mechanism inspired by the SwinTransformer[36]. The encoder-generated embeddings are merged into private and shared modality representations through the factorized fusion mechanism. Private embeddings capture modality-specific information, while shared embeddings encapsulate information common to all modalities. This approach facilitates the learning of cross-modality representations and the association of diverse information available across modalities.

Decoders, also constructed from TS-Transformers, utilize modality embeddings to reconstruct the pre-masking input. Different from previous studies [20, 38], FreqMAE employs a weighted reconstruction objective, leveraging preliminary signal knowledge to prioritize important patches and samples during the pretraining. Specifically, in physical sensing applications, lower-frequency regions with more significant information and signal samples with larger Signal-to-Noise Ratios (SNRs) are prioritized over higher-frequency regions and noisy samples, respectively.

Figure 4: TS-Transformer blocks. Both the Local Window MSA and the TS-Window MSA are multi-head self-attention with local and temporally shifted windows.

### 3.2 Temporal-Shifting (TS) Transformer

The vanilla MAE [20] employs global self-attention within the Transformer, a design well-suited for visual contexts where object semantics are largely independent of their spatial position and scale. Yet, for time-frequency spectrograms, attributes like exact positions, scales, and shifts crucially determine a physical signal's semantics [43]. This creates a misalignment between the vanilla design and our application domain. Figure 2-(a) reveals that while lower frequency band harmonics can predict higher frequency bands vertically, they're less adept at horizontal predictions in the time domain. This is due to higher frequency harmonics shifting gradually from inherent non-stationarity in physical signals. As seen between Figure 2-(a) and (b), this shift complicates predictions using lower frequency bands. The sequence and placement of spectrogram patches are pivotal for signal interpretation. Thus, global attention may be sub-optimal for these spectrograms, especially when time-frequency details are predominantly local.

Inspired by SwinTransformer [36], a state-of-the-art Transformer model for images, TS-Transformer incorporates two fundamental insights: *(i)* the predominantly local time-frequency components of spectrograms, which necessitate an association between local harmonic components, and *(ii)* the need for a representation that captures the shifting frequency components of physical signals due to non-stationarity. Localized attention is essential to ensure limited invariance since (slightly) shifted frequencies resulting from non-stationarity may still represent the same physical phenomenon at different times. Therefore, effective representation learning for physical signals should capture this mechanism while preserving the position and scale of the frequency components.

Figure 4 illustrates the TS-Transformer's design. The masked spectrograms are fed into the patch embedding layer, a convolutional layer that produces a vector embedding from the unmasked patch signals with a dimension of $H_{dim}$. The TS-Transformer consists of two sequential transformer blocks. These blocks take in $H$-dimensional modality embeddings and iterate R times before outputting representations of identical dimensionality. The resulting representation is formulated as:

$$\mathbf{A}_1^{\{r-1\}} = \text{WMSA}\left(\text{LayerNorm}\left(\mathbf{H}^{\{r-1\}}\right)\right) + \mathbf{H}^{\{r-1\}},$$

$$\mathbf{P}^{\{r-1\}} = \text{MLP}\left(\text{LayerNorm}\left(\mathbf{A}_1^{\{r-1\}}\right)\right) + \mathbf{A}_1^{\{r-1\}},$$

$$\mathbf{A}_2^{\{r-1\}} = \text{TS-WMSA}\left(\text{LayerNorm}\left(\mathbf{P}^{\{r-1\}}\right)\right) + \mathbf{P}^{\{r-1\}},$$

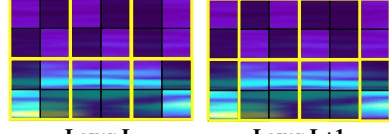

**Layer L**     **Layer L+1**

Figure 5: Local attention and temporal shifted windows.

$$\mathbf{H}^{\{r\}} = \text{MLP}\left(\text{LayerNorm}\left(\mathbf{A}_2^{\{r-1\}}\right)\right) + \mathbf{A}_2^{\{r-1\}},$$

where LayerNorm($\cdot$) is the layer normalization [2]. The MLP($\cdot$) comprises two fully-connected layers. Both WMSA($\cdot$) and TS-WMSA($\cdot$) are multi-head self-attention modules [57] configured with regular (Local Window MSA) and temporally shifted window (TS-Window MSA) attention settings and $A$ attention heads, respectively.

To represent local frequency structures, we employ a *local attention mechanism* for both attention modules. This mechanism only applies attention within short frequency bands while distinctly differentiating across these bands. It groups and segregates spectrogram patches into local windows in spatial dimensions, applying self-attention within these windows to learn relationships among predominantly local frequency components. Furthermore, to address non-stationarity in the temporal dimension, we apply a *temporal shifting* procedure that associates harmonics with their temporally shifted but close counterparts. Figure 5 illustrates the regions of local window attention and the partitioning of temporally shifted windows. The local windows shift horizontally (*i.e.,* in the time dimension) by 50% at consecutive layers to enable cross-window interactions. No shifting is applied to the frequency dimension because different frequency bands carry different physical meanings. This setup facilitates the application of local attention to brief frequency bands to capture primarily local time-frequency components of the spectrogram, while simultaneously recognizing the correlations between shifted harmonics within successive temporal spectrogram windows (*e.g.,* the case in Figure 2).

### 3.3 Factorized Modality Fusion

Multi-modal fusion leverages the diverse and rich information provided by different modalities, each offering a unique perspective on the observed phenomenon. To effectively extract representations from multi-modal data, we emphasize the necessity for *a complementary modality fusion* approach. On one hand, it's vital to *extract shared information between collaborating modalities* to understand their semantic relationships. On the other hand, these modalities mutually enrich each other by offering unique, private information that complements the data from other modalities. A practical framework should be capable of *extracting both shared and unique patterns across modalities to enhance generalizability*.

To achieve this, we introduce a factorized fusion mechanism within FreqMAE, encompassing both modality self-reconstruction and cross-modality reconstruction. Figure 6 provides a visual explanation of this approach. After fusion, each modality's embedding space is partitioned into two subsets: private and shared spaces. Private embeddings come directly from the encoding of the current modality. Conversely, shared embeddings are generated by fusing the embeddings of other modalities through a shared fusion layer, comprising two feed-forward layers. Both private and shared embeddings are then fed into separate decoders to reconstruct the

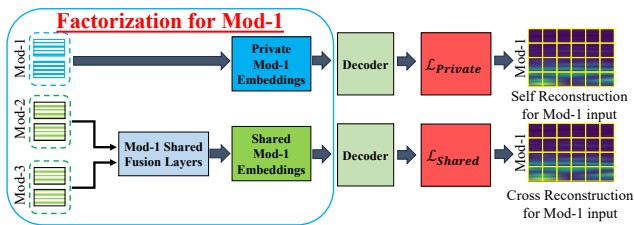

Figure 6: Factorized Fusion in FreqMAE.

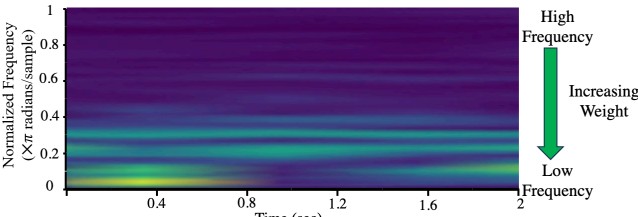

Figure 7: Weighted Mean Square Error weights.

current modality. This reconstruction uses the same weighted loss function, resulting in two distinct reconstruction losses: $\mathcal{L}_{\text{private}}$ and $\mathcal{L}_{\text{shared}}$. The overall pretraining loss is calculated as follows:

$$\mathcal{L}_{\text{total}} = \mathcal{L}_{\text{private}} + \gamma \mathcal{L}_{\text{shared}} \qquad (1)$$

where $\gamma$ is the hyperparameter that controls the weight between two loss components. Because of the asymmetric structure between the deep encoders and lightweight decoders in MAE, we will show later in the evaluation that the introduced extra decoder only incurs negligible computation overhead. Moreover, decoders are discarded after the pretraining stage, removing overhead at inference time.

The proposed factorized fusion mechanism is unique to Freq-MAE and it forces the encoded modality embedding to contain semantical information useful to reconstruct its own input and the input of its peer modalities. In our experiments, we find that a larger $\gamma$ value, which prioritizes shared embeddings, is more effective for datasets with numerous modalities (*e.g.,* IMU data with 3+ modalities). Conversely, a smaller $\gamma$ value proves beneficial for tasks with fewer modalities and heterogeneous information distribution (*e.g.,* an audio-seismic pair, where audio provides richer semantic information). The impact of $\gamma$ is further explored in Appendix D.3. Hence, our fusion scheme is *flexible to accommodate diverse sensor combinations and distributions, with adjustable contributions from private and shared modality information.*

## 3.4 Importance Weighting Loss Function

This module is motivated by two key insights. First, we should emphasize informative content within the signal samples using physical primitives that are common among the sensory data. For instance, in most physical sensing tasks, such as vehicle classification (see Figure 2) and human activity recognition, where the frequency content of most activities lie between 0 and 20 Hz [1], most of the useful information is located in the lower frequency parts of the spectrogram, while high-frequency parts are usually noise [33]. Second, an efficient pretraining objective should emphasize the signal samples containing richer information for the observed physical phenomenon without using labels. Since pretraining is performed with a large amount of unlabeled data, the inherent "class imbalance" is even more evident in such large datasets, where most of the measurements do not contain any activity or context. Devoting excessive attention to reconstructing such samples may cause the model to struggle in capturing meaningful feature patterns.

The vanilla MAE utilizes Mean-Squared Error (MSE) for reconstructing the masked patches during pretraining, defined as:

$$\text{MSE} = \frac{1}{T \times F} \sum_{t=1}^{T} \sum_{f=1}^{F} \left( \mathbf{X}(f,t) - \hat{\mathbf{X}}(f,t) \right)^2, \qquad (2)$$

where $\mathbf{X}$ and $\hat{\mathbf{X}}$ refer to the original and reconstructed spectrograms and $T \times F$ represents the time-frequency dimensionality of the spectrogram. Although it is suitable for images where no preliminary knowledge about object location is known, MSE doesn't perform optimally with sensing spectrogram input. To address this, we leverage our initial insight on prioritizing lower frequency regions, and thus, define the Weighted Mean Squared Error (WMSE) as follows:

$$\text{WMSE} = \frac{1}{T \times F} \sum_{t=1}^{T} \sum_{f=1}^{F} \mathbf{W}_f \left( \mathbf{X}(f,t) - \hat{\mathbf{X}}(f,t) \right)^2, \qquad (3)$$

where $W_f$ refers to the weights of the corresponding spectrogram frequencies. As shown in Figure 7, the weight for the highest frequency is minimum and the weights linearly increase as the frequency decrease. In particular, we set

$$\mathbf{W}_F = \mathbf{W}_{\text{min}}, \qquad \mathbf{W}_1 = \mathbf{W}_{\text{min}},$$

$$\mathbf{W}_f = \mathbf{W}_{\text{max}} - \frac{(f-1)(\mathbf{W}_{\text{max}} - \mathbf{W}_{\text{min}})}{F-1}, \qquad (4)$$

where we set $\mathbf{W}_{\text{min}} = 0$ and $\mathbf{W}_{\text{max}} = 1$ in our experiments.

Besides, in order to prioritize informative samples with movement over background samples, we calculate the mean cumulative energy of the sample across modalities $M$:

$$\text{E} = \frac{1}{M \times T \times F} \sum_{m=1}^{M} \sum_{t=1}^{T} \sum_{f=1}^{F} \mathbf{X}(f,t)^2, \qquad (5)$$

where M is the number of modalities. Note that using the mean cumulative energy across modalities, as opposed to the energies of individual modalities, helps avoid bias towards modalities with typically higher energy content. Since our aim is to comparatively differentiate across samples, the mean energy across modalities provides fair supervision for the training objective. Inspired by the commonly used peak-signal-to-noise ratio (PSNR) metric [21] for comparing image reconstruction quality [24, 48], we define the overall training objective of FreqMAE (in dB) as:

$$WPSNR = 10 \cdot \log\left(\frac{E^\lambda}{WMSE}\right), \qquad (6)$$

where $\lambda$ is the hyperparameter, ranging from 0 to 1, that controls the scale of the energy component. We utilize the negative of WPSNR as the pretraining loss for FreqMAE. Since MSE fundamentally represents the "mean residual energy", both the logarithm in the numerator and the denominator are in the same unit.

The WPSNR objective guides pretraining to prioritize high-fidelity reconstruction of high-energy (low WMSE) samples. In summary, the WPSNR enables the model to emphasize essential frequency

**Table 1: Dataset Summary**

| Dataset | # Classes | Modalities[2] | # Samples | Application |
|---|---|---|---|---|
| MOD | 7 | MP, S | 39,609 | VC |
| ACIDS | 9 | MP, S | 27,597 | VC |
| RealWorld-HAR | 8 | A, G, M, L | 12,887 | HAR |
| PAMAP2 | 18 | A, G, M | 9,611 | HAR |

components within a sample while comparatively assessing the semantic importance of different samples for efficient representation.

## 4 EVALUATION

In this section, we first introduce the experimental setups and then present extensive evaluation results[1] to demonstrate the effectiveness, resiliency, and feasibility of FreqMAE.

### 4.1 Experimental Setup

*4.1.1 Datasets and Preprocessing.* We evaluate FreqMAE's effectiveness with four different datasets used in previous works [10, 42, 55, 58, 66] from two different application domains, *(i)* Vehicle Classification (VC) and *(ii)* Human Activity Recognition (HAR). The datasets cover a comprehensive list of sensors, target classes, sizes, and environments (summarized in Table 1): **(1) MOD** is self-collected using a microphone array and geophone (seismic) to classify moving objects. It has six different vehicle types and a class of human walking. **(2) ACIDS** is collected by the US Army Research Lab for developing acoustic and seismic identification algorithms. It includes 9 different vehicle types in three different terrains. **(3) RealWorld-HAR** is a public dataset collected with an accelerometer, gyroscope, magnetometer, and light sensors. It consists of eight common human activities collected from 15 participants. **(4) PAMAP2** is another public dataset collected via accelerometer, gyroscope, and magnetometers placed on individuals performing 18 different physical activities. More dataset details can be found in Appendix A.

For preprocessing, we partition the time-series data into evenly sized sample windows. We apply the Fourier transform to signals within each interval to generate the spectrum. The sample and interval lengths are determined based on data properties. Resulting spectrograms are fed into FreqMAE to generate feature representations. Note that FreqMAE *can handle different sampling rates among modalities* since they have separate feature encoders.

For training, we randomly divide each dataset into training, validation, and test sets in an 8:1:1 ratio, leaving sessions out to do a realistic split. The training set is further split into different ratios of available labels (100%, 10%, 1%), referred to as **label ratio** during fine-tuning. We evaluate FreqMAE's with low label ratios to *show its effectiveness with scarce data.* More details on preprocessing and training strategies can be found in Appendix B.

*4.1.2 Baselines.* We evaluate FreqMAE against 10 baselines, including a supervised benchmark, five self-supervised representative frameworks that perform instance discrimination (MAE [20], Sim-CLR [7], CAV-MAE [18], AudioMAE [23], LIMU-BERT [65]) two modality-matching based contrastive baselines (CMC [53], Cosmo [39]) and two SOTA contrastive frameworks for time series (TS-TCC

[13], TS2Vec [67]). We provide detailed introductions of baselines in Appendix C. A linear classification layer is appended at the end for downstream tasks during fine-tuning. For the contrastive settings, we keep the backbone encoders the same as FreqMAE for a fair comparison. A set of eight time-domain augmentations, and a frequency domain augmentation is used from common practices [25, 34, 52] for contrastive baselines (augmentations detailed at Appendix B). Note that contrastive frameworks' performance depends on the used augmentations, while *FreqMAE eliminates dependency on used augmentations and is generalizable* (analysis at Section 4.2.1).

### 4.2 Evaluation Results

*4.2.1 Overall Performances.* Table 2 compares the performance of FreqMAE with other baselines using a 100% label ratio. All evaluations use fixed encoders and a linear layer on top of pretrained sample features for a fair assessment of representational quality. The results show FreqMAE surpasses all baselines by at least 6.6 % and 8 % in average accuracy and F1, affirming its effectiveness. While supervised training slightly outperforms FreqMAE on the PAMAP2 task with full labels, we suspect this is due to PAMAP2 including human activities with shorter bandwidth (similar to RealWorld-HAR), therefore self-supervised representations being less detailed to outperform supervised training with full labels. Moreover, supervised training suffers from label shortage and degrades significantly with fewer labels (see Section 4.2.2). Thus, FreqMAE's overall superior performance indicates the high quality of its extracted features. The primary competitors of FreqMAE, TS-TCC and CMC frameworks, are heavily dependent on augmentation design and often underperform with fewer augmentations [61]. Figure 9 demonstrates their performance drop when using only six or three out of nine random augmentations. Further evaluations of FreqMAE on downstream tasks and representation quality are in Appendix D.

*4.2.2 Varying Labeling Ratio.* In this experiment, we evaluate the performances of baselines and FreqMAE with different labeling rates, varying from 1% to 100%. Figure 8 presents the comparison results with all datasets. Higher labeling rates tend to yield improved accuracies across most models. However, FreqMAE consistently outperforms the baseline models in all scenarios. Notably, there are consistent performance gaps between FreqMAE and other models toward lower labeling rates. We note that only TS-TCC consistently competes with FreqMAE. This is because TS-TCC efficiently leverages the temporally correlated nature of sensing signals through temporal contrasting views. However, TS-TCC also relies on a rich set of augmentations and experiences performance degradation with fewer augmentations, as shown in Figure 9. This suggests that FreqMAE *effectively learns general representations from unlabeled data, and thus a linear classifier is enough to achieve higher accuracy.*

*4.2.3 Ablation Study.* Table 3 presents an ablation study using ACIDS for VC and PAMAP2 for HAR tasks to assess the contribution of each design component. We studied four FreqMAE variants: **w/o Weighted Loss** using standard MSE for reconstruction (Equation 2), **w/o Energy Scaling** applying only WMSE loss without energy scaling (Equation 3), **w/o TS-T** employing Swin Transformer instead of TS-Transformer, and **w/o Fusion** without shared fusion and doing separate modality reconstruction during training.

---

[1]Code will be publicly released upon acceptance.
[2]MP=microphone, S=seismic, A=accelerometer, G=gyroscope, L=light, M=magnetometer.

Table 2: Finetune results with 100 % labels. We mark the **best** and second best values.

| Metric | ACIDS | | MOD | | PAMAP2 | | RealWorld-HAR | | Average | |
|---|---|---|---|---|---|---|---|---|---|---|
| | Acc | F1 | Acc | F1 | Acc | F1 | Acc | F1 | Acc | F1 |
| Supervised | 0.9137 | 0.7770 | 0.8948 | 0.8931 | **0.8612** | **0.8384** | 0.9313 | 0.9278 | 0.9002 | 0.8591 |
| CMC | 0.7813 | 0.6216 | 0.9049 | 0.9023 | 0.7571 | 0.7223 | 0.8211 | 0.8384 | 0.8161 | 0.7712 |
| Cosmo | 0.8776 | 0.7298 | 0.3228 | 0.3241 | 0.7910 | 0.7469 | 0.8529 | 0.7968 | 0.7111 | 0.6494 |
| SimCLR | 0.5658 | 0.4879 | 0.7535 | 0.7434 | 0.7346 | 0.6635 | 0.7830 | 0.7181 | 0.7092 | 0.6532 |
| TS2Vec | 0.6539 | 0.4913 | 0.7649 | 0.7632 | 0.5706 | 0.4942 | 0.6117 | 0.5002 | 0.6503 | 0.5622 |
| TS-TCC | 0.9046 | 0.7651 | 0.7709 | 0.7744 | 0.7871 | 0.7107 | 0.8684 | 0.8227 | 0.8328 | 0.7682 |
| Vanilla MAE | 0.8872 | 0.7604 | 0.9015 | 0.8460 | 0.7382 | 0.6999 | 0.8638 | 0.8700 | 0.8477 | 0.7941 |
| LIMU-BERT | 0.5023 | 0.3171 | 0.2157 | 0.1236 | 0.7847 | 0.7612 | 0.7946 | 0.7261 | 0.5743 | 0.4820 |
| CAV-MAE | 0.7995 | 0.6711 | 0.5184 | 0.4941 | 0.7697 | 0.7351 | 0.9215 | 0.9267 | 0.7523 | 0.7068 |
| AudioMAE | 0.7845 | 0.6120 | 0.7274 | 0.7249 | 0.7808 | 0.7478 | 0.8163 | 0.7437 | 0.7773 | 0.7071 |
| FreqMAE | **0.9365** | **0.7919** | **0.9524** | **0.9514** | 0.8420 | 0.8205 | **0.9250** | **0.9327** | **0.9140** | **0.8741** |

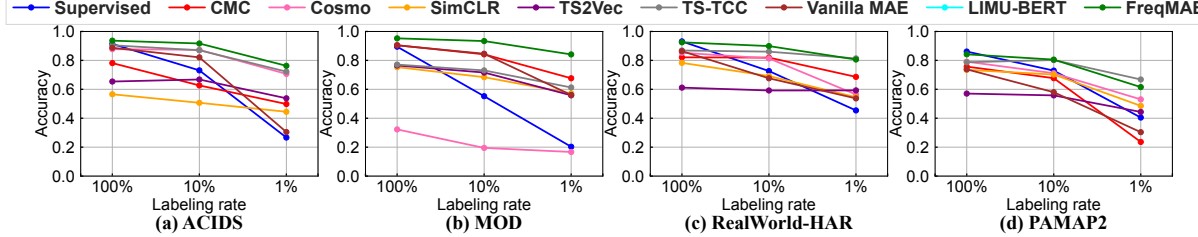

Figure 8: Accuracy comparison of FreqMAE with different labeling rates.

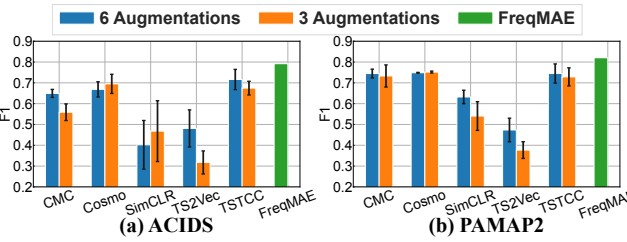

Figure 9: Sensitivity to Data Augmentations.

Table 3: Ablation Study on FreqMAE components.

| Dataset | ACIDS | | PAMAP2 | |
|---|---|---|---|---|
| Metric | Acc | F1 | Acc | F1 |
| w/o Weighted Loss | 0.9068 | 0.7674 | 0.8249 | 0.8046 |
| w/o Energy Scaling | 0.9265 | 0.7642 | 0.8222 | 0.8013 |
| w/o TS-T | 0.9324 | 0.7876 | 0.8238 | 0.7991 |
| w/o Fusion | 0.9183 | 0.7636 | 0.8186 | 0.7905 |
| FreqMAE | **0.9365** | **0.7919** | **0.8420** | **0.8205** |

First, the contribution of all components is evident in both tasks. Comparatively, the fusion component and weighted loss scheme are more helpful in improving task performance, which shows learning relations across modalities can reveal underlying patterns in the frequency domain. Such patterns might be hard to capture without considering modality relations, as different sensor modalities often provide complementary information [40]. Second, the focus of the weighted loss objective on prioritizing informative content within and across samples offers extra self-supervision for pretraining. Finally, the absence of TS-T configuration has a larger impact on the PAMAP2 task than on ACIDS. We suspect this difference is due to the audio and seismic data from the moving vehicles having sparser frequency content with larger temporal correlation (*i.e.,* more stable

Table 4: Compute Overhead Comparison.

| Model | Parameters (M) | Size (MB) | Infer. Time (s) |
|---|---|---|---|
| DeepSense | 0.563 | 2.193 | 0.491 |
| ViT | 2.821 | 10.850 | 1.503 |
| Vanilla MAE | 2.821 | 10.849 | 1.538 |
| FreqMAE | 3.036 | 11.693 | 0.972 |

movement) than HAR tasks. Therefore, the contribution of localized attention and temporal interaction is relatively more limited.

## 4.3 Feasibility in Real-World Deployment

*4.3.1 Computation Overhead.* Table 4 compares FreqMAE with baselines in terms of parameters, model size, and inference time. By running FreqMAE on a single-board Raspberry Pi 3 with 1 GB RAM and a 1.2 GHz quad-core CPU, we evaluate memory and inference time on deployment. The inference time is the execution time for inferring one sample (2-seconds length), averaged over 1000 experiments. Results show that although FreqMAE incurs slightly more inference time than DeepSense [66], a state-of-the-art supervised model for performance comparisons [32, 65], the overhead is comparable and affordable for the considered COTS devices. Moreover, the localized attention mechanism significantly reduces the computational overhead compared to Vanilla MAE, which utilizes a global attention mechanism. Finally, although FreqMAE has comparable size to the ViT, *FreqMAE's local attention mechanism significantly reduces the computational overhead and inference time while improving performance* in sensory data. Hence, *FreqMAE incurs 37% less overhead than its counterparts and allows real-time inference.*

*4.3.2 Robustness Test.* Figure 10 illustrates our field testbed deployment across three distinct parking lot environments: MOD-A, B, and C. We placed FreqMAE sensor nodes with acoustic and seismic

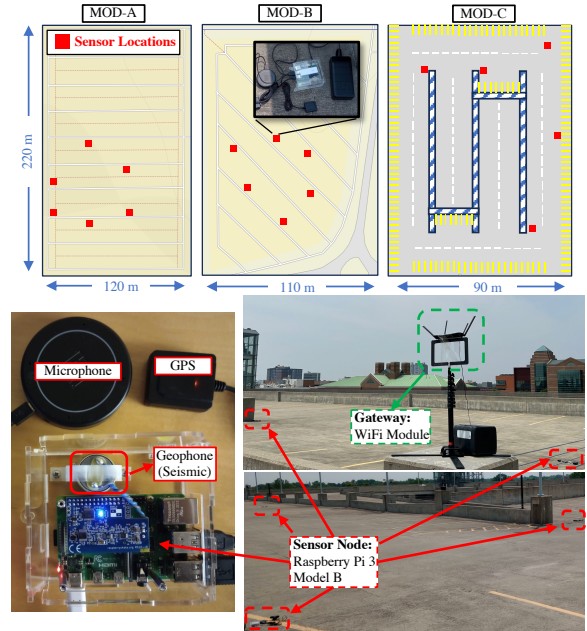

Figure 10: Robustness experiments were conducted in three environments with different variations.

Table 5: MOD variations for robustness experiments.

| Variations | Sensor Locations | Vehicle Types | Terrain | # Labels |
|---|---|---|---|---|
| MOD-A | ✓ | ✗ | ✗ | 3229 |
| MOD-B | ✗ | ✓ | ✗ | 6748 |
| MOD-C | ✗ | ✗ | ✓ | 1163 |

sensors strategically. The pretrained model from the MOD (see Table 1) is utilized for each classification, including variations listed in Table 5. MOD-A aligns closely with the original data, differing only in sensor placement. MOD-B has a similar terrain to MOD-A but uses different vehicles, while MOD-C is set on a concrete building rooftop, introducing distinct acoustic and seismic behaviors.

Table 6 presents the robustness evaluations, *demonstrating Freq-MAE's impressive resilience to environmental variations across deployments*. In MOD-A, changes to sensor locations are less challenging for models, as they mostly influence measurement intensity without significantly altering frequency signatures. For MOD-B, all frameworks struggle with vehicles absent during pretraining due to differing acoustic and seismic signatures with vehicle types. Yet, FreqMAE's performance excels, showcasing its ability to generalize and classify even unseen targets. Finally, in MOD-C, seismic alterations arise due to the concrete environment. However, FreqMAE effectively harnesses insights from physics-based pretraining and the fusion of complementary stable acoustic information, proving adept at distinguishing features even with domain shifts.

Contrastive baselines TS-TCC and CMC, though competitive in standard benchmarks (refer to Table 2 and Figure 8), underperform in changing environments. This drop can be attributed to the nature of contrastive frameworks. While they excel at extracting patterns through similarities among various sample "views", they lack the robustness provided by guidance based on generalized physical features, thereby affecting adaptability in dissimilar environments.

Table 6: Robustness against deployment variations.

| Metric | MOD-A Acc | MOD-A F1 | MOD-B Acc | MOD-B F1 | MOD-C Acc | MOD-C F1 |
|---|---|---|---|---|---|---|
| CMC | 0.7415 | 0.7390 | 0.5760 | 0.4983 | 0.6412 | 0.5691 |
| Cosmo | 0.4205 | 0.3059 | 0.5816 | 0.5214 | 0.5496 | 0.2376 |
| SimCLR | 0.6733 | 0.6685 | 0.5377 | 0.3922 | 0.6107 | 0.3730 |
| TS2Vec | 0.6563 | 0.6439 | 0.5260 | 0.3521 | 0.5725 | 0.4487 |
| TS-TCC | 0.6051 | 0.5910 | 0.5012 | 0.1720 | 0.5802 | 0.4099 |
| Vanilla MAE | 0.8580 | 0.8602 | 0.6626 | 0.6347 | 0.6794 | 0.6326 |
| LIMU-BERT | 0.5000 | 0.1667 | 0.4233 | 0.1983 | 0.5649 | 0.2407 |
| CAV-MAE | 0.4801 | 0.4431 | 0.50309 | 0.21076 | 0.5419 | 0.3409 |
| AudioMAE | 0.5113 | 0.4981 | 0.4839 | 0.3475 | 0.4961 | 0.4571 |
| FreqMAE | **0.8750** | **0.8766** | **0.6885** | **0.6622** | **0.7710** | **0.7340** |

## 5 RELATED WORK

**Self-Supervised Multi-Modal Representation Learning.** Recently, self-supervised learning has progressed in language and vision tasks via contrastive learning [7] and generative models (*e.g.,* MAE) [20]. Early contrastive frameworks focus on instance discrimination, relying on tailored spatial augmentations [7–9, 19]. Multimodal data frameworks, such as CMC [53] and GMC [44], align cross-modality representations without considering frequency structures. Contrastive models tailored for unimodal time series [13, 54, 67–69] exist. Cosmo [39] and Cocoa [11] utilize contrastive learning for multimodal sensing, albeit not optimizing for modality properties. Masked Image Modeling parallels contrastive learning performance in vision [4, 20, 64]. While many have explored Multimodal Modeling, especially for vision-language [15, 29], LIMU-BERT [65] looks at generative modeling for IMU data, but is limited by the sampling rate and does not extend to additional sensory modalities. In contrast, FreqMAE harnesses multimodal traits with shared masked fusion and a physical domain-weighted objective, *enhancing representation learning for multi-modal sensor data.*

**Masked Spectrogram Learning.** MAE, prevalent in vision-based self-supervised learning, is now being applied to Masked Spectrogram Learning [17]. While AudioMAE [23] and MSM-MAE [38] tackle single-modality audio spectrograms, and CAV-MAE [18] blends modality matching with MAE for image and audio, none address the unique characteristics of physical sensory data we motivate. Contrarily, *FreqMAE integrates physical insights in a multimodal approach for enhanced time series representation learning.*

## 6 DISCUSSION AND CONCLUSIONS

The paper introduced an IoT-centric masked autoencoding framework, informed by physics-based insights for sensor signals, to effectively capture crucial semantics for intelligent sensing tasks. Experimental evaluations showed that FreqMAE surpasses current state-of-the-art baselines across different tasks and reduces the need for data labeling, maintaining robustness during domain shifts. A potential limitation of FreqMAE may arise when a significant portion of the unlabeled pretraining data is noisy, potentially affecting the energy supervision from the weighted loss. In such scenarios, adjusting the energy contribution in the training objective to emphasize the reconstruction of important frequency content, typically less noisy, can be beneficial. In future work, we aim to explore training objectives more resilient to such noisy data.

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

# A DATASETS

We evaluate the effectiveness of FreqMAE with four datasets used in previous works [10, 42, 55, 58, 66] from two different applications: *(i)* Vehicle Classification (VC) and *(ii)* Human Activity Recognition (HAR). The datasets cover a comprehensive list of sensors, target classes, sizes, and environments, as detailed in Table 1.

**Moving Object Detection (MOD).** It was independently gathered by us at two locations using a RaspberryShake 1D and a microphone array to record vibration signals from nearby driving automobiles. Seven different object types are involved including a human, at various speeds and distances. The seismic signal was sampled at 100 Hz, whereas the acoustic signal was sampled at 16000 Hz.

**Acoustic-seismic identification Data Set (ACIDS).** It is collected by the US Army Research Lab for training and developing acoustic and seismic identification algorithms. It comprises over 270 data runs from nine different types of ground vehicles in three varying environmental conditions. The data is digitized by a 16-bit A/D at the rate of 1025 Hz.

**RealWorld-HAR [51].** It distinguishes between eight typical human activities, including stair climbing (both up and down), jumping, lying, standing, sitting, running/jogging, and walking, using information from an accelerometer, gyroscope, magnetometer, and light signals. For our experiments, we used data from the waist region collected from 15 participants at a 100 Hz sampling rate.

**PAMAP2 [46].** It incorporates data from 18 diverse physical activities executed by nine individuals using inertial measurement units (IMUs) placed on the chest, wrist (of the dominant arm), and ankle of the dominant side. For our study, we only employed the data recorded from the wrist. Each IMU logs data from a 3-axis accelerometer, gyroscope, and magnetometer, all operating at a sampling rate of 100 Hz.

In the VC application, we use data from varying environments and new vehicle types to create two additional tasks: distance and speed classification. This allows us to evaluate FreqMAE's robustness in the face of domain shifts. For speed classification, the model predicts the vehicle's speed (5, 10, 15, 20 mph), while for distance classification, it identifies if the passing distance is close, mid-range, or far.

# B PREPROCESSING AND TRAINING STRATEGIES

In this section, we provide further details on the datasets, data preprocessing techniques, and training strategies introduced in Section 4.1.

## B.1 Preprocessing.

In the preprocessing phase, we partition the time-series data into evenly-sized sample windows and further divide each sample into either overlapping or non-overlapping intervals. We apply the Fourier transform to signals within each interval to generate the spectrum. The sample and interval lengths are determined based on data properties. Note that FreqMAE can handle different sampling rates among modalities since they have separate feature encoders. Resulting spectrograms are fed into FreqMAE to generate feature representations.

We randomly divide each dataset into training, validation, and test sets in an 8:1:1 ratio, leaving sessions out to do a realistic split. The training set is further split into different ratios of available labels (100%, 10%, 1%), referred to as **label ratio** during finetuning. The unlabeled set is used to perform self-supervised pretraining. In the finetuning phase, decoders are discarded and a linear classifier is trained using the labeled part of the training set and selected by the validation set. Results on the test set are reported.

## B.2 Data Augmentations

In this section, we elaborate on the data augmentation strategies introduced in Section 4.1 for the contrastive baselines. We adopted common practices from previous work when selecting these augmentation strategies to enhance training performance. We categorize the augmentations based on whether they are applied to

time-domain data (time-domain augmentation) or spectrograms (frequency-domain augmentation). *Note that, unlike traditional contrastive frameworks, FreqMAE does not require crafted augmentations for efficient representation learning. It is a self-automated framework capable of generalizing across various IoT task domains.*

*B.2.1 Time Domain Augmentations.* Here, we detail the augmentation strategies used on time series data before converting them into spectrograms.

• **Scaling.** We apply a scaling operation to the input signals by multiplying them with random values drawn from a Gaussian distribution.

• **Permutation.** Within each sample, we introduce a random permutation of intervals.

• **Negation.** We multiply the signal values by a factor of -1.

• **TimeWarp.** We distort the time locations of signal values using a smooth random curve.

• **MagnitudeWarp.** The magnitude of each time series undergoes transformation by multiplication with a curve generated using a cubic spline with randomly positioned knots.

• **Horizontal Flip.** The entire time series of a sample is flipped along the time axis.

• **Jitter.** We introduce random Gaussian noise into the signals.

• **Channel Shuffle.** For multivariate time-series data, such as three-axis accelerometer input (X, Y, Z dimensions), we perform random shuffling of the channels.

*B.2.2 Frequency Domain Augmentations.* Here, we detail the augmentation strategies used on time series data after converting them into spectrograms.

• **Phase Shift.** When dealing with the complex frequency spectrum, we introduce a random phase value within the range of $-\pi$ to $\pi$ to modify their phase values.

## B.3 Training Strategies

In this section, we provide a detailed explanation of the hyperparameters and training strategies employed in the evaluations discussed in Section 4. The specifics of these configurations are tabulated in Table 7. Note that while most of the configurations remain consistent across different backbone encoders, there are slight variations.

Training details and optimization hyperparameters for FreqMAE are presented in Table 8. For the training phase, we utilize the AdamW optimizer paired with cosine schedules. Each framework's initial learning rate is individually tailored based on its unique convergence characteristics. We employ a batch size of 128, and each batch encompasses randomly chosen samples. The temperature parameter is fine-tuned to optimize performance after fine-tuning. A weight decay of 0.05 serves as a regularization strategy throughout training.

In the finetuning stage, we adopt the Adam optimizer coupled with a step scheduler. Specifically, the learning rate diminishes by 0.2 at the end of every epoch. By default, finetuning spans 200 epochs, with each epoch comprising 50 periods. Moreover, we adjust the weight decay for each framework, aiming for the best equilibrium between training and validation fits.

## C  BASELINES

Here, we provide detailed introductions of baselines described in Section 4.1.

• **Supervised.** We train the entire model (*i.e.,* the encoder and linear classifier) in a supervised way with all of the available labels.

• **SimCLR [7].** SimCLR is a robust contrastive learning framework that aims to maximize representation similarity between two randomly augmented views of the same sample while pushing representations of different samples apart. We randomly formulate batches for this work. During pretraining, we generate two distinct views of each sample using random augmentations. SimCLR utilizes a contrastive objective called NT-Xent loss [7] to draw closer to the different transformations of the same samples while pushing away the representations of different samples.

• **CMC [53].** The Contrastive Multiview Coding learns representations by treating representations of the same sample but from different modalities as positive pairs while considering representations of different samples as negative pairs. CMC utilizes the multimodal characteristics of the data to learn meaningful representation. CMC's objective is to maximize the agreement between different modality representations of synchronized data. Each randomly sampled batch with random augmentation leads to the extraction of vector representations for each modality. The system optimizes the backbone parameters by maximizing the similarity between representations of the same samples and treating mismatched modality representations from different samples as negative pairs.

• **MAE [20].** Masked Autoencoder (MAE) is a self-supervised learning approach that leverages the auto-encoding paradigm and the Transformer architecture, achieving state-of-the-art performance in various tasks such as vision [64], audio [38] and robotics [26] tasks. MAE employs a strategy where a large portion of each modality input is randomly masked, and replaced by zeros, ensuring dimensional consistency. It is highly efficient since it only trains on a small portion of unmasked input. Separate encoders and decoders are used for each modality, with the spectrogram transformed into fixed-size patches before the extraction of modality embeddings. Interactions between modalities are facilitated by employing fully connected layers and MLP projection layers on the concatenated modality features. The main objective is to minimize the discrepancy between the original and reconstructed modality patches. For inference, the modality encoders are used to generate latent representations from the unmasked inputs, and a linear layer is applied to these concatenated embeddings for subsequent tasks.

• **LIMU-BERT [65].** LIMU-BERT is a novel representation learning model designed to extract generalized features from unlabeled Inertial Measurement Unit (IMU) data, an abundant and readily available resource. By adopting the self-supervised training principles of BERT, it effectively captures temporal relations and feature distributions in IMU sensor measurements. Despite the original BERT's unsuitability for mobile IMU data, LIMU-BERT successfully adapts to IMU sensing tasks through a series of custom techniques. For a fair comparison, we keep the class head as the original classifier.

• **TS-TCC [13].** It learns robust representations through cross-view predictions and contrasting temporal-contextual information. It generates two views via random data augmentations and predicts

Table 7: TS-Transformer Configurations.

| Dataset | MOD | ACIDS | RealWorld-HAR | PAMAP2 |
|---|---|---|---|---|
| Dropout Ratio | 0.2 | 0.2 | 0.2 | 0.2 |
| Patch Size | aud: [1, 40], sei: [1, 1] | [1, 8] | [1, 2] | [1, 2] |
| Temporal Window Size | [1, 9] | [1, 8] | [1, 9] | [1, 8] |
| Mod Feature Block Num | [2, 2, 4] | [2, 2, 4] | [2, 2, 2] | [2, 2, 2] |
| Mod Feature Block Channels | [64, 128, 256] | [64, 128, 256] | [32, 64, 128] | [32, 64, 128] |
| Head Num | 4 | 4 | 4 | 4 |
| Mod Fusion Channel | 256 | 256 | 128 | 128 |
| Mod Fusion Head Num | 4 | 4 | 4 | 4 |
| Mod Fusion Block | 2 | 2 | 2 | 2 |
| FC Dim | 512 | 512 | 256 | 128 |
| Temporal Shift | 1 | 1 | 1 | 1 |

Table 8: Training configurations. (We use LR for Learning Rate)

| Dataset | MOD | ACIDS | RealWorld-HAR | PAMAP2 |
|---|---|---|---|---|
| Temperature | 0.07 | 0.2 | 0.07 | 0.07 |
| Lambda | 0.1 | 0.3 | 1.0 | 0.3 |
| Gamma | 0.5 | 1.0 | 4.0 | 1.0 |
| Batch Size | 256 | 256 | 256 | 256 |
| Sequence Length | 4 | 4 | 4 | 4 |
| Pretrain Optimizer | AdamW | AdamW | AdamW | AdamW |
| Pretrain Max LR | Default: $1e-5$ | Default: $1e-4$ | Default: $1e-4$ | Default: $1e-4$ |
| Pretrain Max LR | Cosmo, TS2Vec, TSTCC: $1e-5$ | Cosmo: $1e-5$ | CMC: $5e-4$ 
 Cosmo: $1e-5$ | CMC: $5e-4$ 
 Cosmo: $1e-5$ |
| Pretrain Min LR | $1e-07$ | $1e-07$ | $1e-07$ | $1e-07$ |
| Pretrain Scheduler | Cosine | Cosine | Cosine | Cosine |
| Pretrain Epochs | 6000 | 3000 | 1000 | 1000 |
| Pretrain Weight Decay | 0.05 | 0.05 | 0.05 | 0.05 |
| Finetune Optimizer | Adam | Adam | Adam | Adam |
| Finetune Start LR | 0.0001 | 0.0003 | 0.0005 | 0.001 |
| Finetune Scheduler | step | step | step | step |
| Finetune LR Decay | 0.2 | 0.2 | 0.2 | 0.2 |
| Finetune LR Period | 50 | 50 | 50 | 50 |
| Finetune Epochs | 200 | 200 | 200 | 200 |

future timestamps from the context vectors of each view. True future representations are treated as positive pairs, while other sequences are negative pairs. Different augmentations of the same sample are also treated as positive pairs, and different samples within a mini-batch are considered negative pairs.

• **TS2VEC [67].** TS2Vec learns time series representations by iteratively performing temporal and instance contrastive tasks at different sample window sizes. At different granularity, it considers the same sample under various augmentations and sequence contexts as positive pairs, while different samples of both the same and different sequences are treated as negative pairs for instance and temporal contrastive tasks.

• **Cosmo [39].** Cosmo is a framework that leverages contrastive fusion learning to process multimodal time-series data. After encoding and mapping each modality's embedding to a hypersphere, it generates combined features used to calculate contrastive loss, considering similar features as positive pairs and dissimilar ones as negative pairs.

• **AudioMAE [23].** The Audio-MAE framework is introduced as a unified and scalable approach for self-supervised learning of audio

representations. Similar to its predecessor, MAE [20], Audio-MAE employs a Transformer-based encoder-decoder architecture. Unlike MAE, which utilizes global attention during training, AudioMAE **utilizes global and local attention together, making it a good baseline for evaluating the utility of our TS-T design with unique local attention and temporal shift operation**. The process begins by transforming sound into spectrogram patches, of which only a small portion is left unmasked before feeding them into the Transformer encoder for efficient encoding. After padding the encoded patches with learnable embeddings to represent the masked patches, the original order in terms of frequency and time is restored. Subsequently, the data is propagated through a Transformer decoder to reconstruct the audio spectrogram. Unlike image patches, spectrogram patches exhibit significant local correlation, with important information embedded in their frequency and time locations. To address this, localized attention and a hybrid architecture are introduced in the Transformer decoder for improved reconstruction. The primary objective is to minimize the mean

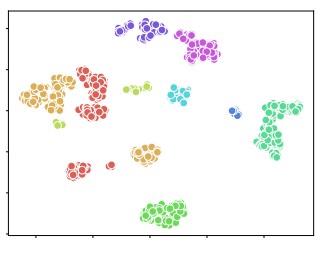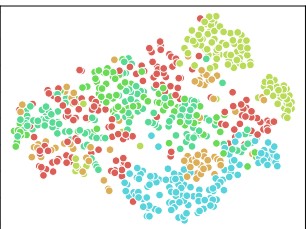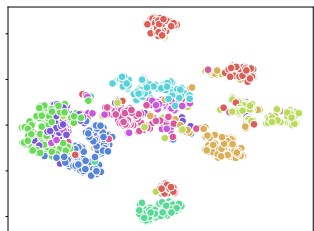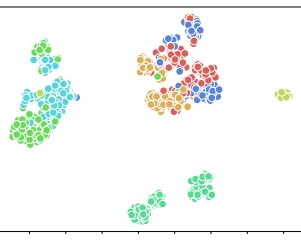

(a) ACIDS  (b) MOD  (c) PAMAP2  (d) RealWorld-HAR

**Figure 11: t-SNE visualization of FreqMAE embeddings. Different colors represent different ground truth labels.**

squared error (MSE) between the predictions and input spectrogram values. In fine-tuning, the decoder is discarded, and the encoder is fine-tuned with patch-masking. Empirically, Audio-MAE achieves state-of-the-art performance on multiple audio and speech classification tasks.

• **CAV-MAE [18].** The Contrastive Audio-Visual Masked Autoencoder (CAV-MAE) is an innovative self-supervised learning method tailored for audio-visual representation. Building upon the principles of the Masked Auto-Encoder (MAE), CAV-MAE extends its capabilities to multi-modal audio-visual contexts and further enriches its performance by introducing contrastive learning. In its essence, the model fuses the concepts of contrastive learning with masked data modeling to generate joint and coordinated audio-visual representations. Through a multi-stream forward pass mechanism, the system ensures the precise separation and combination of audio and visual modalities. A significant portion of the input undergoes masking to facilitate the model's reconstruction efforts, leading to efficient training. For better cross-modal interaction, separate encoders are deployed for each modality, followed by a joint encoder to bridge the modalities. The primary goal is twofold: enforce audio-visual correspondence and accurately reconstruct masked segments. Upon evaluation, CAV-MAE demonstrated its prowess in audio-visual retrieval tasks, setting new benchmarks on the VG-GSound dataset. For real-world applications, the model's encoders generate latent embeddings, which are then processed through a linear layer to derive insightful outcomes for varied tasks.

## D ADDITIONAL EVALUATIONS

### D.1 Additional Downstream Tasks.

We further evaluate the same pretrained models on additional downstream tasks of distance and speed classification using the MOD dataset. The results are presented in Figure 13. Contrastive frameworks for instance-discrimination (*i.e.,* SimCLR), modality-consistency (*i.e.,* CMC), and temporal-contrasting (*i.e.,* TS-TCC) consistently outperform other self-supervised learning methods (*i.e.,,* MAE, LIMU-BERT). FreqMAE integrates modality, temporal characteristics, and physical insights to learn the inherent natures of multimodal time series data, demonstrating superior adaptation on both tasks.

*D.1.1 Representation Visualization.* To evaluate the quality of the representations learned by our model, we apply the t-SNE algorithm [56] to visualize the fused embeddings of FreqMAE. t-SNE algorithm provides a good qualitative benchmark on the distinctive ability of the models through visual representation quality. Figure

11 shows the embedding visualizations constructed by FreqMAE. The t-SNE visualizations reveal distinct and well-separated clusters in ACIDS and RealWorld-HAR datasets, indicating that our model effectively captures the underlying structure of the data. In the case of MOD and PAMAP2, although we observe cohesive clusters, we also notice more overlapped regions, which implies a challenging differentiating structure of the dataset.

### D.2 Effect of Masking Strategies.

We now evaluate FreqMAE's performance under varying masking rates (60 % to 90 %) and strategies, comparing random unstructured masking against three structured variants: *(i)* Time masking for vertical spectrogram patches, *(ii)* Frequency masking for horizontal patches, and *(iii)* Time+Frequency masking, applying both with equal probability. Figure 14 presents the results.

**Masking Rate.** Similar to MAE in vision, we observe that a pre-training high masking ratio (70%-80%) is optimal for sensing spectrogram learning. This is because both images and physics-based signal spectrograms are continuous signals with significant redundancy (see Figure 2). We also found the masking ratio has a bigger effect on the vehicle classification task. This is expected as the audio and seismic data typically have a larger and more complex frequency band, which creates less redundancy and more sensitivity towards the masking ratio than HAR tasks. Moreover, both tasks drop in performance with very high masking ratios (*e.g.,* 90%), presumably because the training task becomes too difficult. This result shows the importance of designing a self-supervised setting with *proper difficulty* in IoT data.

**Masking Scheme.** It's clear that unstructured (random) masking enhances self-supervised pre-training compared to structured approaches. This is due to the model's ability to estimate missing spectrogram components using nearby contexts, such as harmonics in frequency and temporally correlated content in time dimensions. Frequency masking leads to considerable performance degradation due to complete destruction of harmonic bands. Time masking proves more beneficial as missing temporal content can be reconstructed using highly correlated components, enabling the capture of inherent temporal correlation characteristics in physical sensing data. Combining time and frequency masking closely mirrors the performance of unstructured masking due to the ability to extrapolate from nearby content.

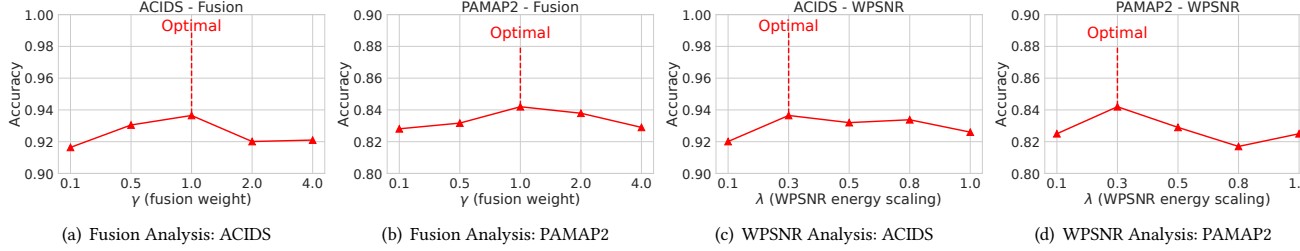

(a) Fusion Analysis: ACIDS    (b) Fusion Analysis: PAMAP2    (c) WPSNR Analysis: ACIDS    (d) WPSNR Analysis: PAMAP2

Figure 12: Fusion ($\gamma$) and WPSNR energy contribution ($\gamma$) hyperparameter anaylsis.

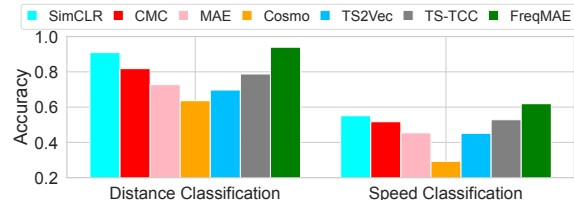

Figure 13: Additional downstream tasks on MOD.

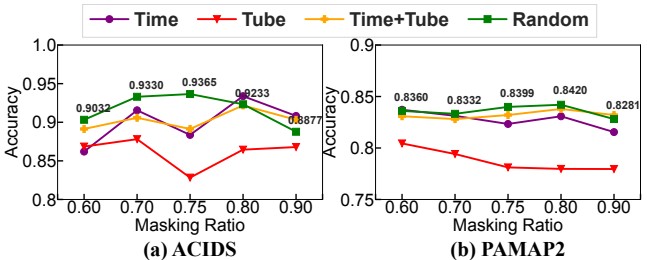

(a) ACIDS        (b) PAMAP2

Figure 14: Effect of masking strategy on performance.

### D.3 Fusion Hyperparameter ($\gamma$) Analysis.

Figure 12-(a, b) illustrates the impact of the information scaling hyperparameter between the shared and private fusion feature embeddings, as discussed in Section 3.3. We evaluated various settings of the hyperparameter ($\gamma$), reporting detection accuracies for two representative datasets (ACIDS and PAMAP2) to assess both VC and HAR tasks. It's important to note that a larger fusion weight emphasizes shared feature characteristics between the modalities. Conversely, a lower value focuses on information derived from individual modalities. Our goal is to determine the optimal fusion settings for different tasks when using FreqMAE.

As illustrated in Figure 12-(a), the ACIDS dataset results indicate that VC tasks often benefit from smaller fusion weights. Given that ACIDS comprises two collaborative modalities—audio and seismic—it is inherently challenging to reconstruct one modality solely from the other. This situation limits the cross-modality information that can be harnessed from the shared fusion embedding space. Moreover, when comparing the seismic and audio modalities, the former lacks the detailed frequency spectrum information present in the latter (as evident in Figure 2). Such an imbalance in spectral content suggests that reconstructing acoustic frequency components solely based on the seismic spectrum might be constrained. Thus, tasks with limited modalities and unbalanced information distribution seem to fare better with reduced fusion weights.

For the PAMAP2 dataset, as shown in Figure 12-(b), which represents HAR tasks, we observe that larger fusion weights generally enhance classification performance. Given that HAR tasks employ IMUs spanning multiple modalities (3+), there are at least two additional modalities available for cross-modality fusion when constructing a shared modality. This abundance enables greater inter-modality collaboration, leading to a richer shared fusion domain.

Overall, our analysis shows that the factorized modality fusion approach offers adaptability for diverse sensor configurations, facilitating the learning of effective representations in mixed modality settings. *Such versatility enables FreqMAE to be applied broadly across various sensing tasks, providing an efficient and generalizable time series data representation framework for practitioners.*

### D.4 WPSNR Hyperparameter ($\lambda$) Analysis.

Figure 12-(c, d) illustrates the influence of energy scaling on the training objective, as detailed in Section 3.4. We explored various energy contribution ($\lambda$) configurations within the loss function, using two exemplar datasets (ACIDS and PAMAP2) to probe both VC and HAR tasks. It's important to highlight that increasing the energy's contribution to the training objective tends to prioritize high-energy samples during training.

Our results indicate that the energy contribution consistently enhances detection performance across tasks. Decreasing the contribution of energy (*i.e.,* $\lambda$) too much tends to reduce the detection performance, as the models start to lose the ability to distinguish between important (*i.e.,* high SNR) samples and samples with no discernable information regarding the observed physical phenomenon (*e.g.,* background data or no observed content).

Moreover, the PAMAP2 analysis on $\lambda$ demonstrates that HAR tasks tend to enjoy higher performance with larger energy contribution (see also the optimal RealWorld-HAR ($\lambda$) configuration in Table 8). This is expected as the detection of the presence of human activities is mostly possible just by looking at the energy content across IMU sensor readings. Thus, the energy contribution to the training objective can exploit this phenomenon and effectively guide the model to learn higher fidelity information from samples with richer information content.

On the other hand, ACIDS results show that setting $\lambda$ too large causes energy contribution to dominate the learned representations. This is not as effective as it is with the HAR tasks, since the IMU sensors for HAR tasks are deployed directly on the human body, and hence, may show less energy variation within an activity of a person [51]. On the contrary, audio and seismic modalities can show a large and swift energy variation for VC tasks [14] since they are deployed outside of faster-moving vehicles. Hence, putting too

**Table 9: The effect of positional encoding on FreqMAE framework**

| Metrics | ACIDS | | PAMAP2 | | RealWorld-HAR | | Parkland | |
|---|---|---|---|---|---|---|---|---|
| | Accuracy | F1 | Accuracy | F1 | Accuracy | F1 | Accuracy | F1 |
| With | 0.9265 | 0.7596 | 0.8312 | 0.8120 | 0.8783 | 0.8916 | 0.9377 | 0.9356 |
| Without | 0.9365 | 0.7919 | 0.8420 | 0.8205 | 0.9250 | 0.9327 | 0.9524 | 0.9514 |

much emphasis on the high-energy samples can result in information loss from when the vehicles are farther away, which still have an audio and seismic signature that provides valuable information.

In summary, our study reveals that the WPSNR training objective with energy supervision enhances task outcomes by guiding the model to focus on high-quality representations. The versatility and effectiveness of the WPSNR objective grant our system adaptability across a range of sensor setups and modality traits. *This adaptability positions FreqMAE as a promising framework to efficiently handle representation tasks across diverse sensing stream configurations.*

## D.5 Effect of Positional Encodings.

In this section, we evaluate the value of positional encoding in the context of the masked representation learning objective. Similar to Swin-Transformers [36], we incorporated absolute positional embeddings (APE) to the embedded patches. Considering that the input spectrogram data varies solely in its temporal dimension, we employ one-dimensional positional embeddings for tokenization. Patch inputs are converted into a one-dimensional sequence, ordered first by channel and then by time, to accommodate variable input lengths. These positional embeddings are subsequently constructed and integrated with the embedding inputs, which then channeled into the backbone network.

Table 9 presents the evaluation results, highlighting the impact of positional encoding on the frameworks. The classification tasks are performed with and without positional embeddings added to the TS-Transformer configuration. Consistent with the findings related to the Swin-Transformer, incorporating positional embeddings into the TS-Transformer offers no evident benefit. In fact, for sensing tasks, it slightly diminishes detection accuracy. We believe this outcome is due to the non-stationary nature of spectrograms. Given that harmonic sequences experience slight temporal shifts with physical primitives (as illustrated in Figure 2), employing positional embeddings as supplementary supervision leads to overfitting on the harmonic sequences specific to each time series sample. Consequently, since the positional information of these harmonics evolves over time, the absolute embedding introduces conflicting guidance, clashing with the temporal relationships that the TS-Transformer deciphers through its distinctive Temporal Shift configuration.

The TS-Transformer groups frequency information within broader localized patches using local attention, thereby encoding inter-frequency relations efficiently for the network. Moreover, the unique *Temporal Shifting* operation enables the learning of shifting harmonics information between patches. Instead of embedding the positions of frequency components directly, this operation fosters attention (and, consequently, association) between the harmonic counterparts of each frequency component. As a result, the TS-Transformer is adept at *efficiently encoding both the relations between frequencies and the non-stationarity phenomenon observed in physical time series data.*

