# OpenReview forum: "FreqMAE: Frequency-Aware Masked Autoencoder for Multi-Modal IoT Sensing"
_ACM.org/TheWebConf/2024/Conference — TheWebConf24_

### Official Review · Reviewer_wSpu · 2023-11-04

**Novelty:** 2
**Technical Quality:** 4

**Review:**

Paper proposes to use masked autoencoder on top of frequency features extracted from raw signals like IMU. The core idea is not new and has been explored extensively e.g., [1] using wavelet transform with contrastive loss. The baselines and tasks are also not very convincing as focus is only on HAR and VC tasks.  Especially, it is unclear how this method performs in comparison with approaches that are SOTA for HAR. The paper's claim regarding being "Multi-Modal" and "IoT Sensing" are very much exaggerated in my opinion not convincing given the tasks used for evaluations.

[1] Saeed, Aaqib, et al. "Federated self-supervised learning of multisensor representations for embedded intelligence." IEEE Internet of Things Journal 8.2 (2020): 1030-1040.

**Questions:**

What new information the method provides in addition to [1] as mentioned above except for using autoencoder with masked input?
How does the method perform if one use MAE on raw signal?
What is the performance of the proposed method in comparison with baseline when one only performs linear probing?
What happens if you train a supervised model with SpecAugment style augmentation? I suspect this will result in a model as good as the FreqMAE in Table 2.
Why results of FreqMAE in Table 2 for RealWorld-HAR are bold when supervised model is performing best?

**Reviewer Confidence:**

4: The reviewer is certain that the evaluation is correct and very familiar with the relevant literature

**Scope:**

3: The work is somewhat relevant to the Web and to the track, and is of narrow interest to a sub-community

---

### Official Review · Reviewer_uDnP · 2023-11-18

**Novelty:** 4
**Technical Quality:** 6

**Review:**

This paper proposes the design of FreqMAE, a self-supervised learning-based framework that uses masked autoencoders for multimodal IoT signals. To achieve a superior performance in comparison to the recent state-of-the-art (SOTA) baselines, FreqMAE distinguish different frequency regions while utilising the cross-modal correlations. Also, FreqMAE defines a weighted loss function that helps it in reconstructing major frequency components that have high SNR.

Strengths:
1. The main strength of this paper is its clear explanation of the broad intuitions and proper reasoning behind the design choices of FreqMAE.
2. The primary methodology is grounded on strong mathematical foundations.
3. The framework has been evaluated on sensor datasets and has been compared to the recent state-of-the-art baselines. These comparisons show the clear advantages that FreqMAE has over the baselines.
4. The paper is highly self-contained and the readers most of the relevant explanations in the draft itself.
5. The paper evaluates FreqMAE on Raspberry Pi 3 and shows that how local attention helps to reduce the computation overload on constrained devices.

Some minor feedback:
1. Although the impact of $\gamma$ is discussed in detail in Appendix D.3, it might be a good idea to include a short summary (2-3 lines) in section 3.3. For example, why a smaller value of $\gamma$ favours systems with fewer modalities.

**Questions:**

No major questions.

**Ethics Review Description:**

No Ethics Issue

**Reviewer Confidence:**

4: The reviewer is certain that the evaluation is correct and very familiar with the relevant literature

**Scope:**

3: The work is somewhat relevant to the Web and to the track, and is of narrow interest to a sub-community

---

### Official Review · Reviewer_hysj · 2023-11-26

**Novelty:** 3
**Technical Quality:** 3

**Review:**

The authors introduce FreqMAE, a self-supervised learning framework designed to enhance feature representation from multi-modal IoT sensing signals. FreqMAE synergizes masked autoencoding with physics-informed signal insights, reducing dependency on data labeling and improving accuracy for AI tasks. Key contributions include a Temporal-Shifting Transformer for time-frequency signal processing, a factorized multimodal fusion mechanism, and a hierarchically weighted loss function that prioritizes important frequency components and high Signal-to-Noise Ratio samples. The framework demonstrates improved performance and resilience against domain shifts in sensing applications, highlighting its potential for foundation models in time-series data analysis.

Overall, the experiments are sufficient and the motivation sounds reasonable. However, there are some weaknesses list below:

The novelty of proposed method is somewhat limited, which is a combination of MAE, Transformer, multi-modal fusion.

The computation efficiency in Table 4 shows the proposed method has more parameters, it is nature to witness higher performance with more parameters.

**Questions:**

The captions for figures need more explanation, such as Fig. 1.

The novelty of proposed method is somewhat limited, which is a combination of MAE, Transformer, multi-modal fusion.

The computation efficiency in Table 4 shows the proposed method has more parameters, it is nature to witness higher performance with more parameters.

**Reviewer Confidence:**

3: The reviewer is confident but not certain that the evaluation is correct

**Scope:**

3: The work is somewhat relevant to the Web and to the track, and is of narrow interest to a sub-community

---

### Official Review · Reviewer_hAXF · 2023-11-26

**Novelty:** 3
**Technical Quality:** 3

**Review:**

The paper introduces FreqMAE, a self-supervised learning framework based on masked autoencoders for multi-modal IoT signals. While demonstrating promising results, the paper lacks clarity in certain areas and could benefit from a more polished writing style.

Pros:

- Novel approach: FreqMAE presents a new self-supervised learning approach by integrating a "physics-informed" module within masked autoencoders.

- Comprehensive evaluations: The extensive evaluations on two sensing applications demonstrate its potential to reduce labeling needs and enhance resilience against domain shifts.

Cons:

- Lack of conceptual clarity: The private and shared concept requires a more detailed motivation and explanation for better understanding.

- Insufficient comparison: The paper falls short in discussing and comparing with reference [68], hindering a comprehensive understanding of its uniqueness in the context of similar ideas.

- writing style issues: Awkwardness and sections that are hard to understand indicate a need for professional editing to enhance overall clarity.

- Term clarification: Certain terms, like "factorization," need clarification to avoid potential misunderstandings.

**Questions:**

Conceptual explanation: Provide a more detailed explanation of the private and shared concepts to improve overall clarity.

Comparison enhancement: Expand the discussion and comparison with reference [68], highlighting both similarities and differences to emphasize the paper's uniqueness.

Professional editing: Consider professional editing to address writing style issues and improve overall clarity, ensuring a smoother reading experience.

Term clarification: Clarify terms such as "factorization" to enhance reader understanding and avoid potential misinterpretations.

**Reviewer Confidence:**

3: The reviewer is confident but not certain that the evaluation is correct

**Scope:**

3: The work is somewhat relevant to the Web and to the track, and is of narrow interest to a sub-community

---

### Official Review · Reviewer_YPSw · 2023-11-27

**Novelty:** 4
**Technical Quality:** 5

**Review:**

The paper presents a masked autoencoder-based framework for evaluating sensing tasks. The authors experiment with multiple datasets and demonstrate the superioroity of their work. The paper is interesting and easy to follow.

The paper is well motivated. The authors have demonstrated the need of the framework in the initial sections. The framework is straightforward, but the use of the frequency domain features helps in the performance. The authors ave explained details of the framework which will make it easy to recreate in future.

The evaluation has been done on multiple publicly avai;able datasets which makes a strong case for the system's performance. The authors have shown how varying the amount of labels does not affect performance. Additionally, the deployment and experimentation on a RPi helps in understanding its robistness

**Questions:**

- There have been deep learing techniques for HAR (e.g., [45] in 2016, and many more over the years). How does this framework perform on the HAR datasets as compared to some of those techniques?

- How does noise affect the data overall? A quick experiment for the same will be helpful.

- What type of datasets are not suitable for the framework?

**Reviewer Confidence:**

3: The reviewer is confident but not certain that the evaluation is correct

**Scope:**

3: The work is somewhat relevant to the Web and to the track, and is of narrow interest to a sub-community

---

### Decision · Program_Chairs · 2024-01-22

**Decision:**

Accept

**Comment:**

The paper addresses a masked autoencoder-based framework for multi-modal IoT signals. The reviewers appreciated the intuition, motivation, and physics-informed module. Two reviewers (uDnP and YPSw) expressed relatively high opinions about the novelty and technical quality, especially on the motivation, mathematical grounding, and comprehensive evaluations. That being said, other reviewers mostly have borderline ratings. In particular, one reviewer (wSpu) criticized that the paper's idea is not necessarily novel with respect to the prior work. Furthermore, the discussion between the authors and the reviewer did not fully resolve the criticism raised.

 Given the evaluations and issues that remain open, I believe this paper is on the fence in terms of acceptability. My AC recommendation is set accordingly.